# LinkerNet: Fragment Poses and Linker Co-Design with 3D Equivariant Diffusion

**Jiaqi Guan**
University of Illinois Urbana-Champaign
jiaqi@illinois.edu

**Xingang Peng**
Peking University
xingang.peng@gmail.com

**Peiqi Jiang**
Tsinghua University
jpq20@mails.tsinghua.edu.cn

**Yunan Luo**
Georgia Institute of Technology
yunan@gatech.edu

**Jian Peng**
University of Illinois Urbana-Champaign
jianpeng@illinois.edu

**Jianzhu Ma**
Tsinghua University
majianzhu@tsinghua.edu.cn

## Abstract

Targeted protein degradation techniques, such as PROteolysis TArgeting Chimeras (PROTACs), have emerged as powerful tools for selectively removing disease-causing proteins. One challenging problem in this field is designing a linker to connect different molecular fragments to form a stable drug-candidate molecule. Existing models for linker design assume that the relative positions of the fragments are known, which may not be the case in real scenarios. In this work, we address a more general problem where the poses of the fragments are *unknown* in 3D space. We develop a 3D equivariant diffusion model that jointly learns the generative process of both fragment poses and the 3D structure of the linker. By viewing fragments as rigid bodies, we design a fragment pose prediction module inspired by the Newton-Euler equations in rigid body mechanics. Empirical studies on ZINC and PROTAC-DB datasets demonstrate that our model can generate chemically valid, synthetically-accessible, and low-energy molecules under both unconstrained and constrained generation settings.

## 1 Introduction

Targeted Protein Degradation (TPD) techniques [27, 7], including PROteolysis TArgeting Chimeras (PROTACs) [35], molecular glues [40], etc., are emerging powerful tools for selectively removing disease-causing proteins. These techniques typically involve multiple molecular fragments connected by a linker, with each fragment binding to a specific protein. For instance, a PROTAC consists of three components: a ligand (warhead) that targets the protein of interest, another ligand that recruits an E3 ubiquitin ligase, and a linker that connects two ligands [38, 9]. PROTACs induce the ubiquitination of target proteins, which is a fundamental biological process where small proteins called ubiquitins are attached to target proteins, marking them for degradation by the proteasome. Unlike traditional small molecule inhibitors that temporarily inhibit protein function by binding to their active sites, PROTAC techniques result in complete protein elimination and offer several advantages, including increased selectivity, reduced off-target effects, and the potential to target previously undruggable proteins [27]. However, designing effective PROTACs remains a significant challenge, particularly in optimizing the linker, which is crucial in maintaining the conformational stability and other important biological properties of the entire molecule.

37th Conference on Neural Information Processing Systems (NeurIPS 2023).

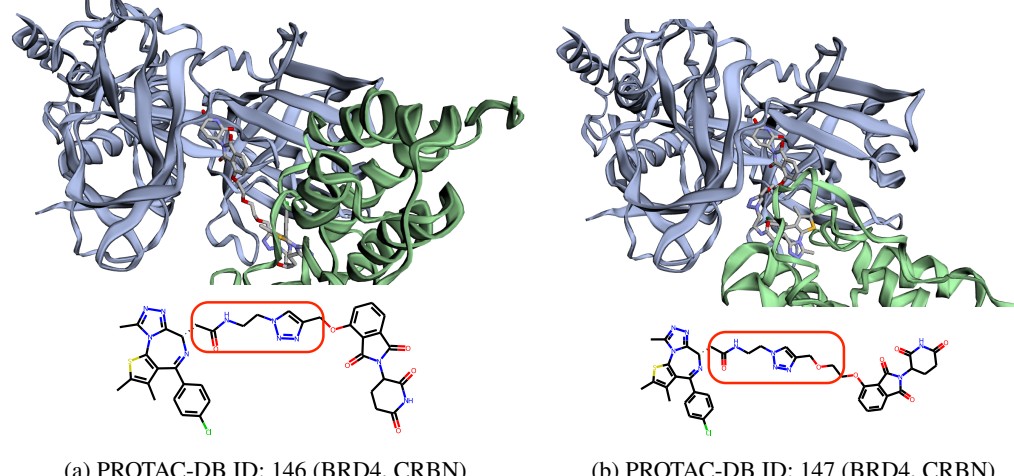

(a) PROTAC-DB ID: 146 (BRD4, CRBN)  (b) PROTAC-DB ID: 147 (BRD4, CRBN)

Figure 1: An example showing that the fragment poses are *not* fixed in PROTAC design. The above sub-figures show two PROTACs, both of which have one fragment binding with the BRD4 target and the other fragment binding with the CRBN E3 ligase. The linkers differ with a 'COC' motif.

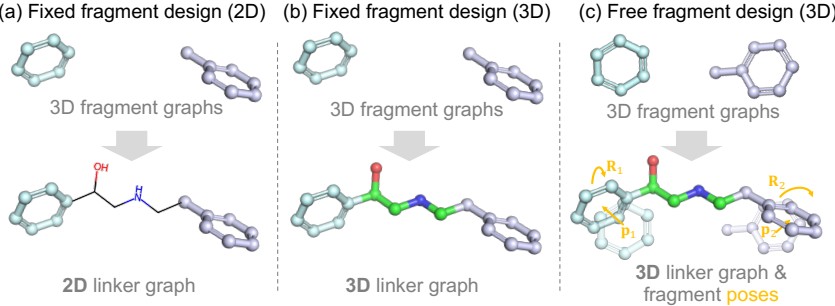

Figure 2: An overview of different linker design settings. (a)/(b): Previous work focuses on 2D/3D linker design with fixed fragment poses. (c): We focus on co-designing fragment poses and linker.

In recent years, computational approaches, particularly deep learning methods, have been employed for linker design to accelerate the drug discovery process [47, 22, 23, 20, 21]. Many of these approaches utilize 3D structural information to enhance their performance. For example, Delinker [22] incorporates the distance and angle between anchor atoms as additional structural information to generate the 2D linker given the fragment molecular graphs. Going a step further, 3DLinker [20] and DiffLinker [21] operate directly in 3D space to process fragments and generate linkers with a conditional VAE and diffusion model, respectively. These models commonly assume that the relative position between fragments is fixed. This assumption is reasonable in traditional fragment-based drug design, where fragments are designed to bind to the same protein pocket and the binding poses are largely deterministic [12]. However, in scenarios involving two proteins (drug target and E3 ligase in the PROTAC case), the relative position between fragments may not be readily available due to the flexibility of protein-protein binding pose, as shown in Figure 1. As a result, it becomes necessary to adjust the fragment poses dynamically during the linker design process. One potential solution is to randomly sample multiple fragment poses and design a 3D linker for each pose. The drawback of this approach is that it limits the design space for the linker by immobilizing the relative positions of the two molecular fragments. It may not be possible to find a stable linker that can connect the two molecular fragments with fixed positions due to the narrow range in which a bond can form. To the best of our knowledge, there is currently no existing computational approach that addresses this challenging 3D linker design problem in the absence of fragment relative positions.

In this work, we first address the problem of co-designing fragment poses and the 3D linker. We represent each fragment as a rigid body and its pose as the position of its center and the rotation. The linker is represented as a 3D graph, consisting of atom positions, atom types, and bond types. To

tackle this co-designing problem, we propose LinkerNet, a 3D equivariant diffusion model, leveraging recent advancements in equivariant diffusion models [19] and diffusion models on Riemann manifolds [8]. Our proposed model can jointly learn the generative process of both fragment poses and the linker, which enables the model to find a stable linker and connect the fragments to form a low-energy conformation of the whole molecule. Moreover, we design a fragment pose prediction module inspired by the Newton-Euler equations in rigid body mechanics. This module employs neural networks to predict atomic forces, which are then used to update the fragment center positions and rotations through the aggregation of neural forces and torques. We introduce two guidance terms that restrict fragment distances and potential anchors to incorporate real-world constraints into our model. To evaluate our method, we perform comprehensive experiments on ZINC [43] and PROTAC-DB [45] datasets and demonstrate that our model can generate chemically valid, synthetically-accessible, and low-energy molecules under both unconstrained and constrained generation settings.

To summarize, our main contributions are:

- Our work presents the first computational model that abandons the unrealistic assumptions in PROTAC drug design.

- We propose a 3D equivariant diffusion model which enables the co-design of fragment poses and the 3D linker structure in a unified framework.

- We develop an effective fragment pose prediction module inspired by the Newton-Euler equations in rigid body mechanics, allowing for the accurate adjustment of fragment center positions and rotations.

- We conduct comprehensive experiments on ZINC and PROTAC-DB datasets, showcasing the superiority of our proposed model over other baseline methods in both unconstrained and constrained generation settings.

## 2  Related Work

**Molecular Linker Design**  Molecular linker design is a critical step in the rational compound design. SyntaLinker [47] operates on the SMILES representation of molecules and formulates the linker design as a sentence completion problem. The lack of 3D structural information and the drawbacks of the SMILES representation itself limits the performance of this method. DeLinker [22] and Develop [23] overcome this limitation by operating on graphs and utilizing the distance and angle between anchor atoms as the additional structural information. However, only limited structural information is used and the generation is still in 2D space. More recently, 3DLinker [20], DiffLinker[21] are proposed to directly generate linkers in 3D space with conditional VAE and diffusion models, respectively. All of these models assume the fragment poses are known. However, this is not always the case, especially in the emerging Targeted Protein Degradation (TPD) techniques. Instead, our model focuses on a more general linker design problem where the fragment poses are unknown.

**PROTAC Linker Design**  PROteolysis TArgeting Chimeras (PROTAC) is a promising technique with many advantages over traditional small molecule inhibitors. The first proof of concept study of PROTAC was proposed in [38]. Most PROTAC linker design strategies rely on empirical optimization of linker composition, which consists of only a few main chemical motifs [44, 32]. Currently, there are no generally accepted rules for *de novo* PROTAC linker design. [48] uses deep reinforcement learning to facilitate rational PROTAC design, but it is still generating SMILES representation instead of molecules in 3D space.

**Diffusion Generative Models**  Diffusion generative models [41, 42, 17] learn to denoise samples from a prior noise distribution and have achieved remarkable progress in generating images [36, 37], text [18, 2], etc. Recently, diffusion models are also applied in molecular data by considering the rotation-translation equivariance, such as molecular conformation generation [46], 3D molecule generation [19] and structure-based drug design [15, 39, 29]. In addition, diffusion models have also been extended to Riemann manifolds [8, 28], and many applications in molecular data have emerged, including conformation generation [25], molecular docking [6], antibody design [31] and protein-ligand binding affinity prediction [24]. Leveraging these advances in equivariance diffusion models and diffusion models in Riemann manifolds, we propose a diffusion model for the fragment

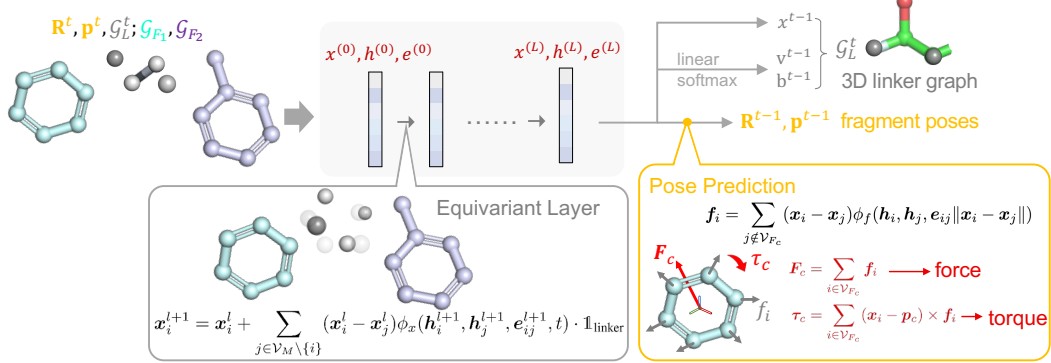

Figure 3: Overview of one denoising step of LinkerNet. An equivariant GNN is applied to update linker atom positions $x$, atom types $v$ and bond types $b$. The updated embedding and positions are then utilized by a pose prediction module to predict neural forces $\boldsymbol{F}_c$ and torques $\boldsymbol{\tau}_c$ to further update fragment poses $\boldsymbol{R}$ and $\boldsymbol{p}$.

poses and linker co-design task. We inject task-specific network and sampling designs, including a physics-inspired fragment poses prediction module and the constrained linker sampling approach.

## 3 Fragment Poses and Linker Co-Design

In this section, we present LinkerNet, which co-designs fragment poses and the linker with 3D equivariant diffusion. We first define notations and this problem formally in Sec. 3.1. Then, we decompose the joint distribution as the product of positions, rotations and atom / bond types in Sec. 3.2 and show how the diffusion processes are constructed for them. In Sec. 3.3, we illustrate the equivariant network and physics-inspired prediction module for denoising fragment poses and the linker. Finally, in Sec. 3.4, we describe how our model can be applied with guided sampling in the constrained generation setting.

### 3.1 Notations and Problem Definition

In our formulation, each molecular fragment is represented as a 3D graph $\mathcal{G}_F = \{\mathbf{v}_F, \mathbf{b}_F, \tilde{\mathbf{x}}_F\}$. We denote the number of atom features, the number of bond types and the number of fragment atoms as $N_a$, $N_b$ and $N_F$, respectively, and then $\mathbf{v}_F \in \mathbb{R}^{N_F \times N_a}$ represents one-hot atom types (including elements and atom charges), $\mathbf{b}_F \in \mathbb{R}^{N_F \times N_F \times N_b}$ represents one-hot bond types (the absence of bond is treated as a special bond type), and $\tilde{\mathbf{x}}_F \in \mathbb{R}^{3N_F}$ is the atom coordinates in the local coordinate system (i.e., conformation). Since we assume fragments are *rigid*, $\tilde{\mathbf{x}}_F$ is unchanged in our setting. We use PCA to construct the local coordinate system [14] for robustness. However, we will see the equivariance of our model is independent of the choice of the local coordinate system (Sec. 3.3). The global pose of each fragment is determined by a rotation transformation $\boldsymbol{R} \in SO(3)$ and a translation transformation $\boldsymbol{p} \in \mathbb{T}(3) \cong \mathbb{R}^3$, i.e. $\mathbf{x}_F = \boldsymbol{R}\tilde{\mathbf{x}}_F + \boldsymbol{p}$. The linker is represented as a set of atom types, bond types and atom positions $\mathcal{G}_L = \{\mathbf{v}_L, \mathbf{b}_L, \mathbf{x}_L\}$, where $\mathbf{v}_L \in \mathbb{R}^{N_L \times N_a}$ denotes linker atom types, $\mathbf{b}_L \in \mathbb{R}^{N_L \times N \times N_b}$ denotes linker bond types and $\mathbf{x}_L \in \mathbb{R}^{3N_L}$ denotes linker atom coordinates in the global coordinate system. Here, $N = N_L + N_{F_1} + N_{F_2}$ is the total number of atoms of two fragments and the linker.

Given two fragments $(\mathcal{G}_{F_1}, \mathcal{G}_{F_2})$ whose global poses are *unknown*, our goal is to design the linker $\mathcal{G}_L$ and recover the global poses of two fragments $(\boldsymbol{R}_1, \boldsymbol{p}_1)$ and $(\boldsymbol{R}_2, \boldsymbol{p}_2)$ to connect fragments with the linker to form a conformationally stable molecule $\mathcal{G}_M$. Specifically, denoting the fragment rotations and translations as $\mathbf{R}$ and $\mathbf{p}$ separately, we aim to learn the distribution $p_\theta(\mathbf{R}, \mathbf{p}, \mathcal{G}_L | \mathcal{G}_{F_1}, \mathcal{G}_{F_2})$ with a neural network parameterized by $\theta$.

### 3.2 Diffusion Processes

A diffusion probabilistic model involves two Markov chains: a forward diffusion process and a reverse generative process. The diffusion process gradually injects noise into data, and the generative process

learns to recover the data distribution from the noise distribution. According to the different types of variables, the joint distribution $p_\theta(\mathbf{R}, \mathbf{p}, \mathcal{G}_L | \mathcal{G}_{F_1}, \mathcal{G}_{F_2})$ can be further decomposed as a product of fragment / linker positions $(\mathbf{x}_L, \mathbf{p})$, fragment rotations $(\mathbf{R})$ and linker atom / bond types $(\mathbf{v}_L, \mathbf{b}_L)$. Next, we will describe how these diffusion processes are constructed in detail.

**Diffusion on Positions**  Denote linker positions $\mathbf{x}_L$ or fragment translations $\mathbf{p}$ as a random variable $\mathbf{x}$. The diffusion on $\mathbf{x}$ involves standard Gaussian diffusion kernels and has been well-studied in [17]. At each time step $t$, a small Gaussian noise is added according to a Markov chain with fixed variance schedules $\beta_1, \ldots, \beta_T$:

$$q(\mathbf{x}_t | \mathbf{x}_{t-1}) = \mathcal{N}(\mathbf{x}_t; \sqrt{1 - \beta_t}\mathbf{x}_{t-1}, \beta_t \mathbf{I}) \tag{1}$$

Under this formulation, we can efficiently draw samples from the noisy data distribution $q(\mathbf{x}_t | \mathbf{x}_0)$ and compute the posterior distribution $q(\mathbf{x}_{t-1} | \mathbf{x}_t, \mathbf{x}_0)$ in closed-form:

$$q(\mathbf{x}_t | \mathbf{x}_0) = \mathcal{N}(\mathbf{x}_t; \sqrt{\bar{\alpha}_t}\mathbf{x}_0, (1 - \bar{\alpha}_t)\mathbf{I}), \qquad q(\mathbf{x}_{t-1} | \mathbf{x}_t, \mathbf{x}_0) = \mathcal{N}(\mathbf{x}_{t-1}; \tilde{\boldsymbol{\mu}}_t(\mathbf{x}_t, \mathbf{x}_0), \tilde{\beta}_t \mathbf{I}), \tag{2}$$

where $\alpha_t = 1 - \beta_t$, $\bar{\alpha}_t = \Pi_{s=1}^t \alpha_s$ and $\tilde{\boldsymbol{\mu}}_t(\mathbf{x}_t, \mathbf{x}_0) = \frac{\sqrt{\bar{\alpha}_{t-1}}\beta_t}{1 - \bar{\alpha}_t}\mathbf{x}_0 + \frac{\sqrt{\alpha_t}(1 - \bar{\alpha}_{t-1})}{1 - \bar{\alpha}_t}\mathbf{x}_t$, $\tilde{\beta}_t = \frac{1 - \bar{\alpha}_{t-1}}{1 - \bar{\alpha}_t}\beta_t$.

**Diffusion on Fragment Rotations**  The diffusion kernel on fragment rotation $\mathbf{R}$ is $\mathcal{IG}_{SO(3)}(\boldsymbol{\mu}, \epsilon^2)$, i.e. the isotropic Gaussian on $SO(3)$ [34, 28] parameterized by a mean rotation $\boldsymbol{\mu}$ and scalar variance $\epsilon^2$. The standard $\mathcal{IG}_{SO(3)}(\mathbf{I}, \epsilon^2)$ can be sampled in an axis-angle form, with uniformly sampled axes $\hat{\boldsymbol{\omega}} \sim \mathfrak{so}(3)$ and rotation angle $\omega \in [0, \pi]$ with density

$$f(\omega) = \frac{1 - \cos\omega}{\pi} \sum_{l=0}^{\infty} (2l + 1)e^{-l(l+1)\epsilon^2}\frac{\sin((l + 1/2)\omega)}{\sin(\omega/2)} . \tag{3}$$

To sample from $\mathcal{IG}_{SO(3)}(\boldsymbol{\mu}, \epsilon^2)$, we can first sample a rotation $e = \omega\hat{\boldsymbol{\omega}}$ and apply it to $\boldsymbol{\mu}$ to obtain the desired sample. Similar to the Euclidean diffusion process, we can also draw samples and compute the posterior distribution at any time step in closed-form:

$$q(\mathbf{R}_t | \mathbf{R}_0) = \mathcal{IG}_{SO(3)}(\lambda(\sqrt{\bar{\alpha}_t}, \mathbf{R}_0), 1 - \bar{\alpha}_t) \quad q(\mathbf{R}_{t-1} | \mathbf{R}_t, \mathbf{R}_0) = \mathcal{IG}_{SO(3)}(\tilde{\boldsymbol{\mu}}_t(\mathbf{R}_t, \mathbf{R}_0), \tilde{\beta}_t), \tag{4}$$

where $\tilde{\boldsymbol{\mu}}_t(\mathbf{R}_t, \mathbf{R}_0) = \lambda\left(\frac{\sqrt{\bar{\alpha}_{t-1}}\beta_t}{1 - \bar{\alpha}_t}, \mathbf{R}_0\right) + \lambda\left(\frac{\sqrt{\alpha_t}(1 - \bar{\alpha}_{t-1})}{1 - \bar{\alpha}_t}, \mathbf{R}_t\right)$ and $\lambda(\gamma, \mathbf{R}) = \exp(\gamma \log(\mathbf{R}))$ denotes the rotation scaling operation, which scales rotation matrices $\mathbf{R}$ by converting them to values in $\mathfrak{so}(3)$, multiplying by a scalar $\gamma$, and converting them back to $SO(3)$ [5].

**Diffusion on Atom and Bond Types**  Following [18], we use categorical distributions to model discrete linker atom types $\mathbf{v}_L$ and bond types $\mathbf{b}_L$. Take atom types as an example (same for bond types), a uniform noise across all $K$ categories is added according to a Markov chain during the diffusion process:

$$q(\mathbf{v}_t | \mathbf{v}_{t-1}) = \mathcal{C}(\mathbf{v}_t | (1 - \beta_t)\mathbf{v}_{t-1} + \beta_t / K). \tag{5}$$

Similarly, we can compute $q(\mathbf{v}_t | \mathbf{v}_0)$ and $q(\mathbf{v}_{t-1} | \mathbf{v}_t, \mathbf{v}_0)$ in closed-forms:

$$q(\mathbf{v}_t | \mathbf{v}_0) = \mathcal{C}(\mathbf{v}_t | \bar{\alpha}_t \mathbf{v}_0 + (1 - \bar{\alpha}_t)/K), \qquad q(\mathbf{v}_{t-1} | \mathbf{v}_t, \mathbf{v}_0) = \mathcal{C}(\mathbf{v}_{t-1} | \tilde{\mathbf{c}}_t(\mathbf{v}_t, \mathbf{v}_0)), \tag{6}$$

where $\tilde{\mathbf{c}}_t(\mathbf{v}_t, \mathbf{v}_0) = \mathbf{c}^\star / \sum_{k=1}^K c_k^\star$ and $\mathbf{c}^\star(\mathbf{v}_t, \mathbf{v}_0) = [\alpha_t \mathbf{v}_t + (1 - \alpha_t)/K] \odot [\bar{\alpha}_{t-1}\mathbf{v}_0 + (1 - \bar{\alpha}_{t-1})/K]$.

### 3.3 Equivariant and Physics-Inspired Neural Network

The likelihood $p_\theta(\mathbf{R}, \mathbf{p}, \mathcal{G}_L | \mathcal{G}_{F_1}, \mathcal{G}_{F_2})$ should be invariant to the global SE(3)-transformation, which can be achieved by composing an invariant initial distribution and an equivariant transition [46, 19]. Thus, we define the distribution on the subspace $\sum_{i=1}^2 \boldsymbol{p}_i = \mathbf{0}$, i.e., the center of fragment positions is zero. It is also consistent with the truth that the linker is around the center of fragments, and thus we can set the prior distribution of $\mathbf{x}_L$ as a standard normal distribution. For the equivariant transition, we model the atomic interaction with a 3D Equivariant GNN $\phi_\theta$:

$$[\hat{\mathbf{x}}_{L,0}, \hat{\mathbf{v}}_{L,0}, \hat{\mathbf{b}}_{L,0}, \hat{\mathbf{R}}_0, \hat{\mathbf{p}}_0] = \phi_\theta(\mathcal{G}_{M_t}, t) = \phi_\theta([\mathbf{x}_{L,t}, \mathbf{v}_{L,t}, \mathbf{b}_{L,t}, \mathbf{R}_t, \mathbf{p}_t], \mathcal{G}_{F_1}, \mathcal{G}_{F_2}, t). \tag{7}$$

To obtain $\mathcal{G}_{M_t}$ from $[\mathbf{x}_t, \mathbf{v}_t, \mathbf{b}_t, \mathbf{R}_t, \mathbf{p}_t]$ and $\{\mathcal{G}_{F_1}, \mathcal{G}_{F_2}\}$, we perform a local-to-global coordinate transformation first and compose the atoms together, i.e., $\mathbf{x}_{F,t} = \mathbf{R}_t \tilde{\mathbf{x}}_{F,t} + \mathbf{p}_t$ and $\mathbf{x}_{M,t} = \{\mathbf{x}_{F,t}, \mathbf{x}_{L,t}\}$. Next, we will elaborate on how the linker and fragment poses are denoised.

**Equivariant Linker Denoising Process** $\phi_\theta$ is a $\mathcal{L}$-layer 3D equivariant GNN. The initial atom and bond embedding $\boldsymbol{h}_i^0$ and $\boldsymbol{e}_{ij}^0$ are obtained by two embedding layers that encode the atom and bond information. At the $l$-th layer, the atom embedding $\boldsymbol{h}_i$, bond embedding $\boldsymbol{e}_{ij}$ and linker atom positions $\boldsymbol{x}_i$ are updated as follows:

$$\tilde{\boldsymbol{e}}_{ij} = \phi_d(\boldsymbol{e}_{ij}^l, \|\boldsymbol{x}_i^l - \boldsymbol{x}_j^l\|) \tag{8}$$

$$\boldsymbol{h}_i^{l+1} = \boldsymbol{h}_i^l + \sum_{j \in \mathcal{V}_M \setminus \{i\}} \phi_h(\boldsymbol{h}_i^l, \boldsymbol{h}_j^l, \tilde{\boldsymbol{e}}_{ij}, t) \tag{9}$$

$$\boldsymbol{e}_{ij}^{l+1} = \boldsymbol{e}_{ij}^l + \sum_{k \in \mathcal{V}_M \setminus \{i\}} \phi_h(\boldsymbol{h}_k^l, \boldsymbol{h}_i^l, \tilde{\boldsymbol{e}}_{ki}, t) + \sum_{k \in \mathcal{V}_M \setminus \{j\}} \phi_h(\boldsymbol{h}_j^l, \boldsymbol{h}_k^l, \tilde{\boldsymbol{e}}_{jk}, t) \tag{10}$$

$$\boldsymbol{x}_i^{l+1} = \boldsymbol{x}_i^l + \sum_{j \in \mathcal{V}_M \setminus \{i\}} (\boldsymbol{x}_i^l - \boldsymbol{x}_j^l) \phi_x(\boldsymbol{h}_i^{l+1}, \boldsymbol{h}_j^{l+1}, \boldsymbol{e}_{ij}^{l+1}, t) \cdot \mathbb{1}_{\text{linker}} \tag{11}$$

where $\mathcal{V}_M$ is the set of all atoms in the molecule and $\mathbb{1}_{\text{linker}}$ is the linker atom mask. The final atom embedding $\boldsymbol{h}_i^{\mathcal{L}}$ and bond embedding $\boldsymbol{e}_{ij}^{\mathcal{L}}$ will be fed into two multi-layer perceptrons and softmax functions to obtain $[\hat{\mathbf{v}}_{L,0}, \hat{\mathbf{b}}_{L,0}]$. In addition, they will also be used to predict the fragment poses $[\hat{\mathbf{R}}, \hat{\mathbf{p}}]$ with a physics-inspired prediction module.

**Physics-Inspired Fragment Pose Prediction** A straightforward way to predict the global pose is by predicting an invariant pose change $(\mathbf{R}_{t \to 0}, \mathbf{p}_{t \to 0})$ in the local coordinate system and applying it to the current pose $(\mathbf{R}_t, \mathbf{p}_t)$ in the global coordinate system, i.e. $(\mathbf{R}_t, \mathbf{p}_t) \circ (\mathbf{R}_{t \to 0}, \mathbf{p}_{t \to 0}) = (\mathbf{R}_t \mathbf{R}_{t \to 0}, \mathbf{p}_t + \mathbf{R}_t \mathbf{p}_{t \to 0})$. This trick has been commonly applied in protein structure prediction [26, 31]. By applying this trick in our setting, it will be applying the invariant pose change prediction from $\boldsymbol{h}_i, \boldsymbol{e}_{ij}$ to the current noisy rotation $\mathbf{R}_t$ and translation $\mathbf{p}_t$ to obtain the denoised ones $\hat{\mathbf{R}}_0$ and $\hat{\mathbf{p}}_0$, i.e.:

$$\hat{\mathbf{R}}_0 = \mathbf{R}_t \phi_R(\boldsymbol{h}_i, \boldsymbol{h}_j, \boldsymbol{e}_{ij}) \qquad\qquad \hat{\mathbf{p}}_0 = \mathbf{p}_t + \mathbf{R}_t \phi_p(\boldsymbol{h}_i, \boldsymbol{h}_j, \boldsymbol{e}_{ij}) \tag{12}$$

However, we argue that the invariant pose update limits the model's capacity since it is regardless of the geometric information of the system in the prediction phase. Considering we treat fragments as *rigid* 3D graphs whose local coordinates will not be changed, it is natural to take inspiration from rigid body mechanics to predict their poses.

The Newton-Euler equations describe a rigid body's combined translational and rotational dynamics. In the Center of Mass (CoM) frame, it can be expressed as the following matrix form:

$$\begin{pmatrix} \boldsymbol{F} \\ \boldsymbol{\tau} \end{pmatrix} = \begin{pmatrix} m\mathbf{I} & 0 \\ 0 & \mathbf{I}_c \end{pmatrix} \begin{pmatrix} d\boldsymbol{v}/dt \\ d\boldsymbol{\omega}/dt \end{pmatrix} + \begin{pmatrix} 0 \\ \boldsymbol{\omega} \times \mathbf{I}_c \boldsymbol{\omega} \end{pmatrix} \tag{13}$$

where $\boldsymbol{F}$ and $\boldsymbol{\tau}$ are the total force and torque acting on CoM, v and $\boldsymbol{\omega}$ are the velocity of CoM and the angular velocity around CoM, $m$ and $\mathbf{I}_c$ is the mass and inertia matrix of the rigid body, which are constant for a given rigid body.

In our fragment pose prediction module, the outputs of the neural network act as the forces $\boldsymbol{f}_i$ on each fragment atom $i$, with which we can compute the total force $\boldsymbol{F}$ and torque $\boldsymbol{\tau}$ for each fragment:

$$\boldsymbol{f}_i = \sum_{j \notin \mathcal{V}_{F_c}} (\boldsymbol{x}_i - \boldsymbol{x}_j) \phi_f(\boldsymbol{h}_i, \boldsymbol{h}_j, \boldsymbol{e}_{ij} \| \boldsymbol{x}_i - \boldsymbol{x}_j \|) \tag{14}$$

$$\boldsymbol{F}_c = \sum_{i \in \mathcal{V}_{F_c}} \boldsymbol{f}_i, \quad \boldsymbol{\tau}_c = \sum_{i \in \mathcal{V}_{F_c}} (\boldsymbol{x}_i - \boldsymbol{p}_c) \times \boldsymbol{f}_i \tag{15}$$

where $c = 1$ or $2$, corresponding to two fragments.

We assume the system is stationary at each discrete time step, i.e. $\boldsymbol{\omega} = 0$ and $\boldsymbol{v} = 0$. Thus, the Newton-Euler equations 13 can be simplified as $\boldsymbol{\tau} = \mathbf{I}_c \frac{d\boldsymbol{\omega}}{dt}$ and $\boldsymbol{F} = m \frac{d\boldsymbol{v}}{dt}$. For a short enough time period $\Delta t$, we have the velocity and angular velocity of the fragment as $\boldsymbol{\omega} = \mathbf{I}_c^{-1} \boldsymbol{\tau}_c \Delta t$ and $\boldsymbol{v} = \frac{1}{m} \boldsymbol{F} \Delta t$. Assuming each atom in the fragment has the unit mass and absorbing the time period $\Delta t$ into $\boldsymbol{F}_c$ and $\boldsymbol{\tau}_c$, the fragment pose will be updated as follows:

$$\hat{\boldsymbol{p}}_{c,0} = \boldsymbol{p}_t + \frac{1}{|\mathcal{V}_{F_c}|} \boldsymbol{F}_c \qquad\qquad \hat{\boldsymbol{R}}_{c,0} = R_\omega(\mathbf{I}_c^{-1} \boldsymbol{\tau}_c) \boldsymbol{R}_t \tag{16}$$

where $|\mathcal{V}_{F_c}|$ denotes the number of atoms in $F_c$ and $R_\omega$ denotes the operation of converting a vector in $\mathfrak{so}(3)$ to a rotation matrix in $SO(3)$ (See Appendix for details).

It can be seen that the predicted force and torque take advantage of the geometric information and are equivariant to the global rigid transformation. Moreover, we also show that the predicted fragment poses are equivariant to the global rigid transformation and are independent of the choices of local coordinate systems. The final training loss is the weighted sum of MSE loss of linker atom positions and fragment center positions, a discrepancy loss of rotation matrix, and KL-divergence of linker atom types and bond types. Please see the proof, more training and modeling details, and the complete training/sampling algorithm in Appendix.

### 3.4 Constrained Generation with Guided Sampling

The LinkerNet we introduced so far is applicable in the case where the fragment poses are fully unknown. However, in real scenarios, we have some prior knowledge, such as the fragment distance should be in a reasonable range, the anchors (the atom on the fragment to connect with the linker) should be selected from a candidate set, etc. To incorporate these constraints, we leverage the idea of classifier guidance [10] to perform guided sampling. In our formulated problem, for a condition $y$, the diffusion score can be modified as:

$$\nabla \log p(\mathcal{G}_{M_t}|y) = \nabla \log p(\mathcal{G}_{M_t}) + \nabla \log p(y|\mathcal{G}_{M_t}). \tag{17}$$

For the fragment distance constraint, we assume the expected distance range is $[d_{\min}, d_{\max}]$, then we have the following guidance term:

$$-\nabla_{\mathbf{p}} \max(\|\boldsymbol{p}_1 - \boldsymbol{p}_2\| - d_{\max}, 0) + \max(d_{\min} - \|\boldsymbol{p}_1 - \boldsymbol{p}_2\|, 0). \tag{18}$$

To generate a complete and valid molecule, at least one atom from the candidate anchor set should form a bond with one linker atom, and the atoms outside the anchor set should not form bonds with any linker atoms. These principles can be formulated as the following anchor proximity guidance:

$$-\nabla_{\mathbf{R}} \max(d_a - r_{\max}, 0) + \max(r_{\min} - d_a, 0) + \max(r_{\max} - d_{na}, 0). \tag{19}$$

where $\mathcal{A}$ is the candidate anchor set, $d_a = \min_{i \in \mathcal{V}_{\mathcal{G}_L}, j \in \mathcal{A}} \|\boldsymbol{x}_i - \boldsymbol{x}_j\|$ and $d_{na} = \min_{i \in \mathcal{V}_{\mathcal{G}_L}, j \in \mathcal{V}_{\mathcal{G}_F} \backslash \mathcal{A}} \|\boldsymbol{x}_i - \boldsymbol{x}_j\|$. $r_{\min}$ and $r_{\max}$ denote the minimum and maximum of a bond length, which are set to 1.2Å and 1.9Å in practice. Besides the soft constraint on anchor proximity, we can also set a hard constraint by applying a bond mask during the sampling phase, i.e. we adopt $\mathbf{b}_L \in \mathbb{R}^{N_L \times |\mathcal{A}| \times N_b}$ instead of $\mathbb{R}^{N_L \times N \times N_b}$

## 4 Experiments

### 4.1 Setup

We mainly conduct experiments in two settings: *unconstrained* generation and *constrained* generation. In the unconstrained generation setting, only 3D fragment graphs $\mathcal{G}_{F_1}, \mathcal{G}_{F_2}$ are known, and the goal is to validate whether the model can co-design linker and fragment poses to generate molecules $\mathcal{G}_M$ with low-energy and other desired properties. In the constrained generation setting, the candidate anchor set $\mathcal{A}$ on each fragment is also known besides $\mathcal{G}_{F_1}, \mathcal{G}_{F_2}$, and we set a fragment center distance constraint. The goal is to validate whether the model can perform well in a more realistic scenario. We describe the general setting about data, baselines, and evaluation metrics as follows and the detailed task-specific settings in the corresponding subsections.

**Data**  We use a subset of ZINC [43] for the unconstrained generation. Following [22, 20, 21], the reference conformation for each molecule is obtained by running 20 times MMFF [16] optimization using RDKit [1] and selecting the one with lowest energy. We use the same procedure as [21] to create fragments-linker pairs and randomly split the dataset, which results in a training/validation/test set with 438,610 / 400 / 400 examples. For the constrained generation, we use PROTAC-DB [45], a database collecting PROTACs from the literature or calculated by programs. The same procedure is applied to obtain reference conformations and create data pairs. We select 10 different warheads as the test set (43 examples) and the remaining as the training set (992 examples).

Table 1: Unconstrained generation results on ZINC. (*) denotes additional anchor information is utilized and thus scores are not directly comparable. (++) denotes a huge number (>100k in energy). The mean and standard deviation values are reported by running the sampling procedure 3 times with different random seeds.

| Method | Valid,% | Unique,% | Novel,% | Rec,% | QED ($\uparrow$) | SA ($\downarrow$) | $E_{\min}$ ($\downarrow$) | RMSD ($\downarrow$) | $E_L$ ($\downarrow$) | $\Delta E_L$ ($\downarrow$) |
|---|---|---|---|---|---|---|---|---|---|---|
| DeLinker | **96.8 ± 0.2** | 43.5 ± 0.4 | 43.3 ± 0.2 | 55.8 ± 1.2 | 0.61 ± 0.0 | 3.13 ± 0.0 | - | - | - | - |
| 3DLinker | 40.3 ± 0.1 | 53.6 ± 0.6 | 47.4 ± 2.7 | 43.2 ± 0.9* | 0.55 ± 0.00 | 3.08 ± 0.00 | ++ | 2.42 ± 0.01 | 5178.2 ± 108.5 | ++ |
| DiffLinker | 48.7 ± 0.1 | **90.1 ± 3.5** | **99.5 ± 0.0** | 0.0 | 0.59 ± 0.00 | 7.25 ± 0.00 | 179.2 ± 6.7 | 1.92 ± 0.00 | 216.8 ± 1.0 | 93.3 ± 0.4 |
| Ours | 83.1 ± 0.0 | 14.8 ± 0.1 | 11.4 ± 0.1 | 24.4 ± 0.1 | **0.70 ± 0.00** | **3.01 ± 0.00** | **32.7 ± 1.2** | **1.44 ± 0.00** | **54.8 ± 10.5** | **67.9 ± 1.3** |

**Baselines** For benchmarking, we compare our model with three baselines: DeLinker [22], 3DLinker [20] and DiffLinker [21]. DeLinker is a 2D graph generative model, while 3DLinker and DiffLinker are 3D generative models with VAE and diffusion models, respectively. Since there is no existing generative model to perform the fragment poses and linker co-design, we randomly sample fragment rotations and add noise to fragment center positions. Then, the noisy fragments are fed to these models to generate linkers.

**Evaluation Metrics** For ZINC and PROTAC-DB dataset, we generate 250 and 100 samples per fragment pair respectively for further evaluation. We evaluate the generated molecules on both 2D graphs and 3D conformations. For 2D metrics, we report the standard ones including validity, uniqueness and novelty [4], the recovery rate (the percentage of generated molecules that can successfully recover the reference molecules), and property-related metrics drug-likeness (QED) [3] and synthetic accessibility (SA) [13]. To evaluate 3D conformations, we first perform MMFF [16] optimization, and report the average minimum energy of generated molecules per fragment pair *before* optimization as $E_{\min}$, and the average Root Mean Square Deviation (RMSD) of the molecule coordinates before and after optimization as RMSD. $E_{\min}$ indicates the best quality of the overall generated conformations and RMSD indicates the gap between generated conformations and the best possible ones. To further investigate the model's performance in generating linker and fragment poses respectively, we perform another constrained MMFF optimization by fixing the atoms in fragments. Then, we report the average median energy *after* optimization as $E_L$ and the average median energy difference before and after optimization as $\Delta E_L$. Since fragment atoms are fixed, the optimization will fix the unrealistic conformation inside the linker and thus a lower $E_L$ indicates the better fragment poses, and a lower $\Delta E_L$ means less adjustment on linker atoms and thus indicates the better linker conformation.

## 4.2 Unconstrained Generation Results

For baseline models, we randomly sample fragment rotations and add a Gaussian noise on the fragment distance (stddev = 0.1 distance). We filter out clashing initial fragments and feed each baseline with 250 valid initial fragment poses per fragment pair. We fix the number of linker atoms to be the same as the reference molecule. For DeLinker and 3DLinker baselines, anchor atoms are also provided to utilize their published models while they are unavailable for DiffLinker and our model.

From Table 1, we can first see that our model can generate much more valid molecules compared to other 3D linker generative models (3DLinker and DiffLinker). Although our model's uniqueness and novelty scores are lower than other baselines, we believe that is because the chemical linker design space to form a low-energy molecule is limited. Our model achieves a recovery rate of $24.5\%$. In contrast, DiffLinker achieves zero recovery rate since it can not co-design fragment poses, while DeLinker / 3DLinker utilizes the anchor information and thus achieve a higher recovery rate. In terms of QED and SA, our model clearly outperforms other models, which indicates our generated linkers are more realistic. For 3D-related evaluation metrics, our model can achieve much lower energy and RMSD compared to other baselines, which justifies the effectiveness of our modeling approach.

Since we argue that our physics-inspired fragment pose prediction module is more effective in Sec. 3.3, we conduct an ablation study (See Table 2) on it to further investigate the role of each part. We take equation 12 as the counterpart of the Newton-Euler equations-based update formula (equation 16). Firstly, we find the fragment translation prediction based on Newton equation is more effective than predicting the change of translation in the local frame. The latter approach will result in severe drift (the fragments are far away from each other) during the sampling phase, as reflected in

Table 2: Ablation study on the fragment pose prediction module.

| | Pose Pred Newton | Pose Pred Euler | Valid, % | Rec, % | $E_{\min}$ (↓) | RMSD (↓) | $E_L$ (↓) | $\Delta E_L$ (↓) |
|---|---|---|---|---|---|---|---|---|
| (a) | | | 81.1 | 47.3 | ++ | ++ | ++ | ++ |
| (b) | ✓ | | **97.2** | **70.8** | 180.4 | 2.22 | 2052.2 | ++ |
| (c) | | ✓ | 81.6 | 20.3 | ++ | ++ | ++ | ++ |
| (d) | ✓ | ✓ | 83.1 | 24.5 | **32.2** | **1.44** | **49.3** | **67.1** |

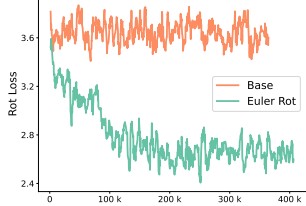

Figure 4: Rotation loss.

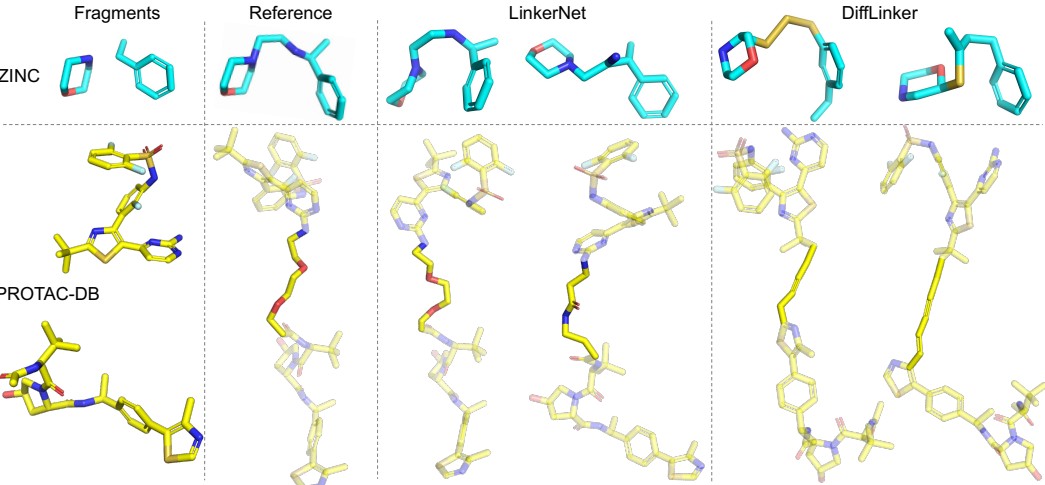

Figure 5: Visualization of reference molecules and molecules generated by LinkerNet and DiffLinker.

the huge energy and RMSD in (a) and (c), even though the model can learn to connect one linker atom with each fragment to achieve reasonable scores in 2D metrics such as validity and recovery rate. Secondly, we can see the fragment rotation prediction based on Euler equation (d) can achieve lower energy and RMSD compared to its counterpart (b), indicating our design can boost the model's capacity and make more accurate rotation predictions. We further plot the rotation training loss in Fig. 4, which confirms our argument since the rotation loss decreases as expected with Euler-equation rotation prediction while it almost remains unchanged with the counterpart design.

### 4.3 Constrained Generation Results

In the constrained generation setting, we mimic the real scenario in the PROTAC linker design. Since two fragments also need to bind the protein of interest and E3 ligase separately, the possible anchor can only be chosen from a subset of the fragment atoms. In addition, the linker length has a critical effect on the PROTAC's selectivity, and we usually need to restrict the range of fragment distance. To include these constraints, we take the atoms within two hops of the real anchor atom as the imaginary candidate anchor atom set for each fragment. We set the fragment center distance constraint as $[0.8d, 1.2d]$, where $d$ is the fragment distance in the reference molecule.

We generate samples with guided sampling and hard bond mask as described in 3.4. For baselines, the fragment distance is uniformly sampled within the constraint, and random anchors from candidate anchor sets are provided. The number of linker atoms is sampled according to the approach proposed in their original paper. Since the public pretrained 3DLinker model can not support the linker generation to large molecular graphs such as PROTACs, we do not include it for comparison.

Table 3 shows the constrained generation results on PROTAC-DB. Our model achieves a higher validity and recovery rate, and outperforms other baselines in three of 3D metrics ($E_{\min}$, RMSD and $E_L$) with a clear margin. DiffLinker has a lower energy difference before and after linker force filed optimization than ours, which indicates molecules generated by DiffLinker has better geometry inside the linker. It makes sense since DiffLinker focuses on learning the linker distribution only, while our model is trained on a more complex fragment poses and linker co-design task.

Table 3: Constrained generation results on PROTAC-DB. The mean and standard deviation values are reported by running the sampling procedure 3 times with different random seeds.

| Method | Valid,% | Unique,% | Novel,% | Rec,% | $E_{\min}$ ($\downarrow$) | RMSD ($\downarrow$) | $E_L$ ($\downarrow$) | $\Delta E_L$ ($\downarrow$) |
|---|---|---|---|---|---|---|---|---|
| DeLinker | $42.8 \pm 0.4$ | $89.9 \pm 2.8$ | $99.0 \pm 0.1$ | $0.0 \pm 0.0$ | - | - | - | - |
| DiffLinker | $24.0 \pm 0.1$ | $\mathbf{99.4 \pm 0.0}$ | $\mathbf{98.9 \pm 0.2}$ | $0.0 \pm 0.0$ | $416.2 \pm 13.4$ | $2.44 \pm 0.05$ | $501.0 \pm 17.6$ | $\mathbf{80.8 \pm 4.6}$ |
| Ours | $\mathbf{55.5 \pm 0.0}$ | $47.9 \pm 11.6$ | $41.4 \pm 4.9$ | $\mathbf{5.4 \pm 1.4}$ | $\mathbf{113.7 \pm 8.0}$ | $\mathbf{1.55 \pm 0.02}$ | $\mathbf{29.9 \pm 17.1}$ | $500.0 \pm 99.1$ |

Table 4: Ablation study on the guided sampling.

| | Guidance | | Valid,% | DistSucc,% | $E_{\min}$ ($\downarrow$) | RMSD ($\downarrow$) | $E_L$ ($\downarrow$) | $\Delta E_L$ ($\downarrow$) |
| | Anchor | Distance | | | | | | |
|---|---|---|---|---|---|---|---|---|
| (a) | | | 45.9 | 25.2 | ++ | 1.66 | 124.3 | ++ |
| (b) | ✓ | | 53.2 | 25.4 | 1249.7 | 1.61 | 432.5 | 1495.0 |
| (c) | | ✓ | 49.3 | **53.7** | 208.4 | 1.56 | 45.4 | 4971.0 |
| (d) | ✓ | ✓ | **55.5** | 52.6 | **115.6** | **1.54** | **18.2** | **610.7** |

To justify the effect of each guidance term, we conduct another ablation study as shown in Tab. 4. Without guided sampling, the model will generate unrealistic 3D linkers for some fragment pairs and result in a very large average energy $E_{\min}$ and $\Delta E_L$. Anchor proximity guidance (b) and fragment distance (c) are both of benefit for meeting the constraints and achieving lower energy and RMSD. The combination of them (d) achieves the best result.

## 5 Conclusion

We introduce LinkerNet for 3D fragment poses and linker co-design, the first work which addresses this more general and challenging linker design task. The limitation of our model is that it does not directly incorporate fragment rotation constraints or explicitly consider the protein context in modeling. These aspects could be valuable directions for future research.

**Reproducibility Statements** The model implementation, experimental data and model checkpoints can be found here: `https://github.com/guanjq/LinkerNet`

**Acknowledgement** We thank all the reviewers for their feedbacks through out the review cycles of the manuscript. This work was supported by the National Key Research and Development Program of China grants 2022YFF1203100 and 2021YFF1201600

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

## A  Exponential and Logarithmic Mapping between $\mathfrak{so}(3)$ and $SO(3)$

A rotation matrix has an associated axis-angle representation. The transformation between them relies upon the exponential and logarithmic mapping between the Lie algebra $\mathfrak{so}(3)$ and $SO(3)$.

Following standard definitions [5], the logarithm of a rotation matrix $\boldsymbol{R}$ is:

$$\log(\boldsymbol{R}) = \frac{\theta}{2\sin\theta}(\boldsymbol{R} - \boldsymbol{R}^T) \,, \tag{20}$$

where $\text{Tr}(\boldsymbol{R}) = 1 + 2\cos\theta$. It can be shown that the logarithm of $\boldsymbol{R} \in SO(3)$ is a skew-symmetric matrix $\boldsymbol{S} \in \mathfrak{so}(3)$:

$$\boldsymbol{S} := \log(\boldsymbol{R}) = \begin{pmatrix} 0 & -v_z & v_y \\ v_z & 0 & -v_x \\ -v_y & v_x & 0 \end{pmatrix} \,, \tag{21}$$

where $\boldsymbol{v} = [v_x, v_y, v_z]$ is the rotation axis and $\theta = \|\boldsymbol{v}\|_2$ is the rotation angle.

Correspondingly, the exponential of a skew-symmetric matrix is a rotation matrix:

$$\boldsymbol{R} := \exp(\boldsymbol{S}) = \boldsymbol{I} + \frac{\sin\|\boldsymbol{v}\|_2}{\|\boldsymbol{v}\|_2}\boldsymbol{S} + \frac{1 - \cos\|\boldsymbol{v}\|_2}{\|\boldsymbol{v}\|_2^2}\boldsymbol{S}^2 \,. \tag{22}$$

Following [28], we diffuse the rotation matrix by scaling the angle of rotation along the geodesic from the identity, which can be done by logarithmic mapping the rotation matrix to values in $\mathfrak{so}(3)$, element-wise multiplying by a scalar, and exponentially mapping them back to $SO(3)$, i.e. $\lambda(\gamma, \mathbf{R}) = \exp(\gamma \log(\mathbf{R}))$. We follow the same way as [31] to pre-compute and cache discretized angle distribution to draw samples efficiently.

## B  Proof of Equivariance

### B.1  Equivariance w.r.t Global SE(3)-Transformation

We denote the global SE(3)-transformation as $T_g$, which can be written explicitly as $\boldsymbol{x}' = T_g(\boldsymbol{x}) = \boldsymbol{R}_g\boldsymbol{x} + \boldsymbol{b}$, where $\boldsymbol{R}_g \in \mathbb{R}^{3\times3}$ is the rotation matrix and $\boldsymbol{b} \in \mathbb{R}^3$ is the translation vector. Applying $T_g$ will result in the same transformation on fragment poses, i.e. $\boldsymbol{p}'_c = \boldsymbol{R}\boldsymbol{p}_c$, $\boldsymbol{R}'_c = \boldsymbol{R}_g\boldsymbol{R}_c$.

First, considering the zero-CoM (Center of Mass) operation $\bar{\boldsymbol{x}}_i = \boldsymbol{x}_i - (\boldsymbol{p}_1 + \boldsymbol{p}_2)/2$, the translation vector in the global SE(3)-transformation will be cancelled out:

$$\bar{\boldsymbol{x}}'_i = T_g(\bar{\boldsymbol{x}}) = \boldsymbol{R}\boldsymbol{x}_i + \boldsymbol{b} - (\boldsymbol{R}\boldsymbol{p}_1 + \boldsymbol{b} + \boldsymbol{R}\boldsymbol{p}_2 + \boldsymbol{b})/2 = \boldsymbol{R}(\boldsymbol{x}_i - (\boldsymbol{p}_1 + \boldsymbol{p}_2)/2) = \boldsymbol{R}\bar{\boldsymbol{x}}_i \,. \tag{23}$$

Thus, we only need to consider the rotation transformation. Next, we will prove the equivariance in the linker denoising process and fragment pose prediction.

**Linker Denoising Process**  It is easy to see the atomic distance $\|\boldsymbol{x}_i - \boldsymbol{x}_j\|$ is $T_g$. Thus, $\tilde{\boldsymbol{e}}_{ij}, \boldsymbol{h}_i, \boldsymbol{e}_{ij}$ are also invariant since the update of them (as shown in Eq. (8, 9, 10)) only involves invariant inputs. For $\tilde{\boldsymbol{x}}_i$, the update formula is

$$\phi(\boldsymbol{x}_i^l) = \boldsymbol{x}_i^l + \sum_{j\in\mathcal{V}_M\backslash\{i\}} (\boldsymbol{x}_i^l - \boldsymbol{x}_j^l)\phi_x(\boldsymbol{h}_i^{l+1}, \boldsymbol{h}_j^{l+1}, \boldsymbol{e}_{ij}^{l+1}, t) \cdot \mathbb{1}_{\text{linker}} \,. \tag{24}$$

After applying $T_g$, we have

$$
\begin{aligned}
\phi(T(\boldsymbol{x}_i^l)) &= T(\boldsymbol{x}_i^l) + \sum_{j\in\mathcal{V}_M\backslash\{i\}} (T(\boldsymbol{x}_i^l) - T(\boldsymbol{x}_j^l))\phi_x(\boldsymbol{h}_i^{l+1}, \boldsymbol{h}_j^{l+1}, \boldsymbol{e}_{ij}^{l+1}, t) \cdot \mathbb{1}_{\text{linker}} \\
&= \boldsymbol{R}\boldsymbol{x}_i^l + \sum_{j\in\mathcal{V}_M\backslash\{i\}} \boldsymbol{R}(\boldsymbol{x}_i^l - \boldsymbol{x}_j^l)\phi_x(\boldsymbol{h}_i^{l+1}, \boldsymbol{h}_j^{l+1}, \boldsymbol{e}_{ij}^{l+1}, t) \cdot \mathbb{1}_{\text{linker}} \\
&= \boldsymbol{R}\left( \boldsymbol{x}_i^l + \sum_{j\in\mathcal{V}_M\backslash\{i\}} (\boldsymbol{x}_i^l - \boldsymbol{x}_j^l)\phi_x(\boldsymbol{h}_i^{l+1}, \boldsymbol{h}_j^{l+1}, \boldsymbol{e}_{ij}^{l+1}, t) \cdot \mathbb{1}_{\text{linker}} \right) \\
&= T(\phi(\boldsymbol{x}_i^l)) \,,
\end{aligned}
\tag{25}
$$

which implies that the linker atom position update is equivariant. By stacking multiple layers together, we can draw the conclusion that the denoised linker atom positions are SE(3)-equivariant.

**Fragment Poses Prediction**   We recap the fragment poses prediction as follows:

$$
\boldsymbol{f}_i = \sum_{j \notin \mathcal{V}_{F_c}} (\boldsymbol{x}_i - \boldsymbol{x}_j) \phi_f(\boldsymbol{h}_i, \boldsymbol{h}_j, \boldsymbol{e}_{ij} \| \boldsymbol{x}_i - \boldsymbol{x}_j \|)
$$

$$
\boldsymbol{F}_c = \sum_{i \in \mathcal{V}_{F_c}} \boldsymbol{f}_i, \quad \boldsymbol{\tau}_c = \sum_{i \in \mathcal{V}_{F_c}} (\boldsymbol{x}_i - \boldsymbol{p}_c) \times \boldsymbol{f}_i \tag{26}
$$

$$
\hat{\boldsymbol{p}}_{c,0} = \boldsymbol{p}_t + \frac{1}{|\mathcal{V}_{F_c}|} \boldsymbol{F}_c, \quad \hat{\boldsymbol{R}}_{c,0} = R_\omega(\mathbf{I}_c^{-1} \boldsymbol{\tau}_c) \boldsymbol{R}_t
$$

where $c = 1$ or $2$, corresponding to two fragments.

First, it is easy to see $\boldsymbol{f}_i$ is equivariant w.r.t $T_g$ following the similar proof about the equivariance of linker atom positions. Thus, we can prove that the total force $\boldsymbol{F}_c$ and torque $\boldsymbol{\tau}_c$ are equivariant:

$$
\boldsymbol{F}_c' = \sum_{i \in \mathcal{V}_{F_c}} \boldsymbol{R}_g \boldsymbol{f}_i = \boldsymbol{R}_g \sum_{i \in \mathcal{V}_{F_c}} \boldsymbol{f}_i = \boldsymbol{R}_g \boldsymbol{F}_c ,
$$

$$
\boldsymbol{\tau}_c' = \sum_{i \in \mathcal{V}_{F_c}} \boldsymbol{R}_g(\boldsymbol{x}_i - \boldsymbol{p}_c) \times \boldsymbol{R}_g \boldsymbol{f}_i \tag{27}
$$

$$
= \boldsymbol{R}_g \sum_{i \in \mathcal{V}_{F_c}} (\boldsymbol{x}_i - \boldsymbol{p}_c) \times \boldsymbol{f}_i = \boldsymbol{R}_g \boldsymbol{\tau}_c .
$$

Second, we follow the same inertia matrix definition as [24]: $\mathbf{I}_c = \sum_{i \in \mathcal{V}_{F_c}} \| \boldsymbol{x}_i - \boldsymbol{p}_c \|^2 \mathbf{I} - (\boldsymbol{x}_i - \boldsymbol{p}_c)(\boldsymbol{x}_i - \boldsymbol{p}_c)^\top$. After applying $T_g$, we have $\mathbf{I}_c' = \boldsymbol{R}_g \mathbf{I}_c \boldsymbol{R}_g^{-1}$, and thus the angular velocity is equivariant:

$$
\boldsymbol{\omega}' = \mathbf{I}_c'^{-1} \boldsymbol{\tau}_c' = \boldsymbol{R}_g \mathbf{I}_c^{-1} \boldsymbol{R}_g^{-1} \boldsymbol{R}_g \boldsymbol{\tau}_c = \boldsymbol{R}_g \boldsymbol{\omega} \tag{28}
$$

Third, we notice one nice property of Lie group is that the adjoint transformation is linear:

$$
\mathbf{R} \exp(\boldsymbol{\omega}) = \exp(\mathrm{Adj}_{\mathbf{R}} \boldsymbol{\omega}) \mathbf{R} , \qquad \text{for } \boldsymbol{\omega} \in \mathfrak{so}(3), \mathbf{R} \in SO(3) \tag{29}
$$

In the case of SO(3), the adjoint transformation for an element is exact the same rotation matrix used to represent the element [11], i.e. $\mathrm{Adj}_{\mathbf{R}} = \mathbf{R}$. Thus, we have $\mathbf{R} \exp(\boldsymbol{\omega}) = \exp(\mathbf{R} \boldsymbol{\omega}) \mathbf{R}$.

As a result, the predicted fragment center positions are equivariant:

$$
\hat{\boldsymbol{p}}_{c,0}' = \boldsymbol{R}_g \boldsymbol{p}_t + \frac{1}{|\mathcal{V}_{F_c}|} \boldsymbol{R}_g \boldsymbol{F}_c = \boldsymbol{R}_g \hat{\boldsymbol{p}}_{c,0} , \tag{30}
$$

and the predicted fragment rotations are also equivariant:

$$
\hat{\boldsymbol{R}}_{c,0}' = \exp(\boldsymbol{R}_g \boldsymbol{\omega}) \boldsymbol{R}_g \boldsymbol{R}_t = \boldsymbol{R}_g \exp(\boldsymbol{\omega}) \boldsymbol{R}_g^{-1} \boldsymbol{R}_g \boldsymbol{R}_t = \boldsymbol{R}_g \exp(\boldsymbol{\omega}) \boldsymbol{R}_t = \boldsymbol{R}_g \hat{\boldsymbol{R}}_{c,0} . \tag{31}
$$

## B.2   Independence w.r.t Local Coordinate System

Since we always take the fragment center as the origin of the local coordinate system, we only need to consider the effect of the orientation change. Suppose the global coordinates remain unchanged, and the change of local coordinate system results in a rotation $\boldsymbol{R}_c$ applied on local coordinates, i.e. $\tilde{\mathbf{x}}' = \boldsymbol{R}_s \tilde{\mathbf{x}}$. Thus, the transformation between local coordinates and global coordinates becomes $\mathbf{x} = \boldsymbol{R} \boldsymbol{R}_s^{-1} \tilde{\mathbf{x}}' + \boldsymbol{p}$, which implies the new rotation matrix representing the fragment pose is $\boldsymbol{R} \boldsymbol{R}_s^{-1}$.

Recall that the fragment rotation update is $\hat{\boldsymbol{R}}_0 = R_\omega(\mathbf{I}_c^{-1} \boldsymbol{\tau}_c) \boldsymbol{R}_t$. With the change of local coordinate system, the predicted fragment rotation is $\hat{\boldsymbol{R}}_0' = R_\omega(\mathbf{I}_c^{-1} \boldsymbol{\tau}_c) \boldsymbol{R}_t \boldsymbol{R}_s^{-1}$. Since our rotation loss is defined as $L_{\mathrm{rot}} = \| \boldsymbol{R}_0 \hat{\boldsymbol{R}}_0^\top - \mathbf{I} \|_F^2$, we have

$$
L_{\mathrm{rot}}' = \| \boldsymbol{R}_0 \boldsymbol{R}_s^{-1} (\hat{\boldsymbol{R}}_0 \boldsymbol{R}_s^{-1})^\top - \mathbf{I} \|_F^2 = \| \boldsymbol{R}_0 \boldsymbol{R}_s^{-1} \boldsymbol{R}_s \hat{\boldsymbol{R}}_0^\top - \mathbf{I} \|_F^2 = L_{\mathrm{rot}} , \tag{32}
$$

which means the training of our model is independent w.r.t the choice of local coordinate system.

## C   Implementation Details

### C.1   Featurization

The molecular graph is extended as a fully-connected graph. The atom features include a one-hot element and charge indicator (H, C, N, $N^-$, $N^+$, O, $O^-$, F, Cl, Br, I, S(2), S(4), S(6)) and a one-hot fragment/linker indicator. Note that the fragment/linker can be predetermined. Thus, it will only serve as the input feature without getting involved in the network's prediction. The edge features include a one-hot bond type indicator (None, Single, Double, Triple, Aromatic), and a 4-dim one-hot vector indicating the edge is between fragment atoms, linker atoms, fragment-linker atoms or linker-fragment atoms.

### C.2   Model Details

Atom features and edge features are firstly fed to two embedding layers with `node_emb_dim=256` and `edge_emb_dim=64`. The hidden embeddings get involved in three types of layers: atom update layer, bond update layer, and position update layer as described in Eq. (9, 10, 11). In each layer, we concatenate the input features and update the hidden embedding / positions with a 2-layer MLP with LayerNorm and ReLU activation. The stack of these three layers is viewed as a block, and our model consists of 6 blocks. For the force prediction layer, we apply graph attention to aggregate the message of each node/edge. The key/value/query embedding is also obtained with a 2-layer MLP.

We set the number of diffusion steps as 500. For this diffusion noise schedule, we choose to use a cosine $\beta$ schedule suggested in [33] with `s=0.01`.

### C.3   Loss Functions

For the linker atom positions and fragment center positions losses, we use the standard mean Squared Error (MSE). For the rotation loss, we measure the discrepancy between the real and the predicted rotation matrices by computing the Frobenius-norm of $\|\boldsymbol{R}_0\hat{\boldsymbol{R}}_0^\top - \mathbf{I}\|$. For the atom and bond type losses, we compute the KL divergence between the real posterior and the predicted posterior. Specifically, the loss functions can be summarized as follows:

$$L_{\text{linker}} = \|\mathbf{x}_L - \hat{\mathbf{x}}_{L,0}\|^2 \tag{33}$$

$$L_{\text{tr}} = \Sigma_{c=1}^2 \|\boldsymbol{p}_c - \hat{\boldsymbol{p}}_{c,0}\|^2 \tag{34}$$

$$L_{\text{rot}} = \Sigma_{c=1}^2 \|\boldsymbol{R}_{c,0}\hat{\boldsymbol{R}}_{c,0}^\top - \mathbf{I}\|_F^2 \tag{35}$$

$$L_{\text{atom}} = \sum_{k=1}^{N_a} \tilde{\boldsymbol{c}}(\mathbf{v}_{L,t}, \mathbf{v}_{L,0})_k \log \frac{\tilde{\boldsymbol{c}}(\mathbf{v}_{L,t}, \mathbf{v}_{L,0})_k}{\tilde{\boldsymbol{c}}(\mathbf{v}_{L,t}, \hat{\mathbf{v}}_{L,0})_k} \tag{36}$$

$$L_{\text{bond}} = \sum_{k=1}^{N_b} \tilde{\boldsymbol{c}}(\mathbf{b}_{L,t}, \mathbf{b}_{L,0})_k \log \frac{\tilde{\boldsymbol{c}}(\mathbf{b}_{L,t}, \mathbf{b}_{L,0})_k}{\tilde{\boldsymbol{c}}(\mathbf{b}_{L,t}, \hat{\mathbf{b}}_{L,0})_k} \tag{37}$$

The final loss is a weighted sum of them:

$$L = \lambda_1 L_{\text{linker}} + \lambda_2 L_{\text{tr}} + \lambda_3 L_{\text{rot}} + \lambda_4 L_{\text{atom}} + \lambda_5 L_{\text{bond}} \tag{38}$$

### C.4   Training Details

The model is trained via AdamW [30] with `init_learning_rate=5e-4`, `betas=(0.99, 0.999)`, `batch_size=64` and `clip_gradient_norm=50.0`. To balance the scales of different losses, we multiply a factor $\lambda = 100$ on the atom type loss and bond type loss. During the training phase, we add a small Gaussian noise with a standard deviation of 0.05 to linker atom coordinates as data augmentation. We also schedule to decay the learning rate exponentially with a factor of 0.6 and a minimum learning rate of 1e-6. The learning rate is decayed if there is no improvement for the validation loss in 10 consecutive evaluations. The evaluation is performed for every 2000 training steps. We trained our model on one NVIDIA RTX A6000 GPU, and it could converge within 350k steps.

### C.5 Overall Training and Sampling Procedures

In this section, we summarize the overall training and sampling procedures of LinkerNet as Algorithm 1 and Algorithm 2, respectively.

---

**Algorithm 1** Training Procedure of LinkerNet

---

**Input:** Linker dataset $\{\mathbf{R}, \mathbf{p}, \mathcal{G}_L, \mathcal{G}_{F_1}, \mathcal{G}_{F_2}\}_{i=1}^N$, where $\mathcal{G}_L = \{\mathbf{v}_L, \mathbf{b}_L, \mathbf{x}_L\}$, $\mathcal{G}_F = \{\mathbf{v}_F, \mathbf{b}_F, \tilde{\mathbf{x}}_F\}$ and $\boldsymbol{p}_1 + \boldsymbol{p}_2 = 0$; Neural network $\phi_\theta$

1: **while** $\phi_\theta$ not converge **do**
2:     Sample diffusion time $t \in \mathcal{U}(0, \ldots, T)$
3:     Add noise to $(\mathbf{x}_L, \mathbf{p})$, $\mathbf{R}$ and $(\mathbf{v}_L, \mathbf{b}_L)$ according to Eq. (2, 4, 6), respectively
4:     Compose $[\mathbf{x}_t, \mathbf{v}_t, \mathbf{b}_t, \mathbf{R}_t, \mathbf{p}_t]$ to form $\mathcal{G}_{M_t}$ by performing $\mathbf{x}_{F,t} = \mathbf{R}_t \tilde{\mathbf{x}}_{F,t} + \mathbf{p}_t$
5:     Move the molecule to make $\boldsymbol{p}_{1,t} + \boldsymbol{p}_{2,t} = \mathbf{0}$
6:     Predict $[\hat{\mathbf{x}}_{L,0}, \hat{\mathbf{v}}_{L,0}, \hat{\mathbf{b}}_{L,0}, \hat{\mathbf{R}}_0, \hat{\mathbf{p}}_0]$ from $\mathcal{G}_{M_t}$ with $\phi_\theta$ as described in Sec. 3.3
7:     Compute the training loss $L$ according to Eq. (38)
8:     Update $\theta$ by minimizing $L$
9: **end while**

---

---

**Algorithm 2** Sampling Procedure of LinkerNet

---

**Input:** The molecular fragments $\mathcal{G}_{F_1}, \mathcal{G}_{F_2}$, the learned model $\phi_\theta$. Optional: the fragment distance constraint $[d_{\min}, d_{\max}]$, the fragment candidate anchor sets $\mathcal{A}$.
**Output:** Generated fragment poses $(\mathbf{R}, \mathbf{p})$ and linker $\mathcal{G}_L$

1: If the fragment distance constraint is provided, sample the number of atoms in $\mathcal{G}_L$ based on a prior distribution summarized from the training set.
2: Sample initial fragment poses $(\mathbf{R}_T, \mathbf{p}_T)$ and linker $\mathcal{G}_{L,T}$
3: **for** $t$ in $T, T-1, \ldots, 1$ **do**
4:     Compose $[\mathbf{x}_t, \mathbf{v}_t, \mathbf{b}_t, \mathbf{R}_t, \mathbf{p}_t]$ to form $\mathcal{G}_{M_t}$ by performing $\mathbf{x}_{F,t} = \mathbf{R}_t \tilde{\mathbf{x}}_{F,t} + \mathbf{p}_t$
5:     Move the molecule to make $\boldsymbol{p}_{1,t} + \boldsymbol{p}_{2,t} = \mathbf{0}$
6:     Predict $[\hat{\mathbf{x}}_{L,0}, \hat{\mathbf{v}}_{L,0}, \hat{\mathbf{b}}_{L,0}, \hat{\mathbf{R}}_0, \hat{\mathbf{p}}_0]$ from $\mathcal{G}_{M_t}$ with $\phi_\theta$ as described in Sec. 3.3
7:     Sample $[\hat{\mathbf{x}}_{L,t-1}, \hat{\mathbf{v}}_{L,t-1}, \hat{\mathbf{b}}_{L,t-1}, \hat{\mathbf{R}}_{t-1}, \hat{\mathbf{p}}_{t-1}]$ from the posterior according to Eq. (2, 4, 6)
8:     Optional: Compute the guidance according to Eq. (18, 19) if the corresponding constraint is provided. Update the prediction with guidance according to Eq. (17).
9: **end for**

---

## D  Training and Sampling Efficiency

For the training efficiency, DiffLinker converges within 300 epochs and takes 76 hrs with one V100 GPU as the original paper reported. Our model converges within 50 epochs and takes 48 hrs with the same type of GPU.

For the sampling complexity, DiffLinker finished sampling linkers for 43 PROTAC fragment pairs (100 linkers for each pair) in 132 min while our model takes 761 min with the same NVIDIA 1080 GPU.

