# OpenReview forum: "LinkerNet: Fragment Poses and Linker Co-Design with 3D Equivariant Diffusion"
_NeurIPS.cc/2023/Conference — NeurIPS 2023 spotlight_

### Official Review · Reviewer_ZLo5 · 2023-06-17

**Soundness:** 3 good
**Presentation:** 3 good
**Contribution:** 2 fair
**Rating:** 6
**Confidence:** 4

**Summary:**

In this paper, the authors formulated a new linker design task where the fragment poses are unknown. The authors proposed a 3D equivariant diffusion model which enables the co-design of fragment poses and the linker structure in a unified framework.

**Strengths:**

1. The paper is well-written and easy to follow.
2. The authors developed an effective fragment pose prediction module inspired by the Newton-Euler equations in rigid body mechanics, allowing for the accurate adjustment of fragment center positions and rotations. This is the main technical contribution compared with the previous work DiffLinker.
3.  Comprehensive experiments on ZINC and PROTAC-DB datasets demonstrate the superiority of LinkerNet over other baseline methods in both unconstrained and constrained generation settings.

**Weaknesses:**

1. The authors do not show the application of LinkerNet for molecule generation conditioned on the target protein.
2. The code is not provided.

**Questions:**

please see the weakness.

**Limitations:**

The authors have adequately discussed the limitations.

---

> ### Author Rebuttal · Authors · 2023-08-09
>
> We thank the reviewer for the feedback and suggestions. Please see below for our responses to the comments.
>
> **Q1: The authors do not show the application of LinkerNet for molecule generation conditioned on the target protein.**
>
> A1:  Thank you for pointing this out! This is indeed one limitation as we discussed in the Conclusion section. We will consider the protein context and apply our model in a more realistic scenario in future work.
>
> **Q2: The code is not provided.**
>
> A2: We are committed to open-sourcing the data, training/inference code, and the model checkpoint once the paper is published.

---

> > ### Comment · Reviewer_ZLo5 · 2023-08-15
> > **Thanks for the rebuttal**
> >
> > I have read the reply and appreciate the author's reply. My concerns are mostly resolved. Thanks!

---

### Official Review · Reviewer_4AKh · 2023-06-25

**Soundness:** 3 good
**Presentation:** 3 good
**Contribution:** 3 good
**Rating:** 6
**Confidence:** 3

**Summary:**

The article discusses the problem of designing linkers to connect different molecular fragments in order to form stable drug-candidate molecules, specifically in targeted protein degradation techniques such as PROteolysis TArgeting Chimeras (PROTACs). One significant challenge in these techniques is that existing models for linker design assume that the relative positions of the fragments are known, which may not be the case in real scenarios. This problem is addressed by the authors through the development of a 3D equivariant diffusion model that jointly learns the generative process of both fragment poses and the 3D structure of the linker, viewing fragments as rigid bodies and designing a fragment pose prediction module inspired by the Newton-Euler equations in rigid body mechanics.

The proposed 3D equivariant diffusion model, called LinkerNet, jointly learns the generative process of both fragment poses and the 3D structure of the linker. Fragments are viewed as rigid bodies and designed with a fragment pose prediction module inspired by the Newton-Euler equations in rigid body mechanics. To address the problem of designing linkers when fragment poses are unknown in 3D space, LinkerNet is able to co-design fragment poses and the linker. The model represents each fragment as a rigid body and its pose as the position of its center and the rotation. The linker is a 3D graph with atom positions, atom types, and bond types. LinkerNet is a diffusion model for the fragment poses and linker co-design task, involving an equivariant network and physics-inspired prediction module for denoising fragment poses and the linker. The model is able to generate chemically valid, synthetically-accessible, and low-energy molecules under both unconstrained and constrained generation settings.

**Strengths:**

First, the proposed method, LinkerNet, differs from previous methods in linker generation by co-designing fragment poses and the linker in a 3D equivariant diffusion model, instead of assuming that the relative positions of the fragments are known. Previous methods mostly focus on 2D or 3D linker design with fixed fragment poses, while LinkerNet addresses the more general linker design problem where fragment poses are unknown in 3D space. This perspective is new and can be a good guidance for future works on linker design.

The authors also introduce a physics-based neural network for their task, which is also novel. The physics-inspired neural network in LinkerNet is advantageous in co-designing fragment poses and the 3D linker for stable drug-candidate molecules. It allows for predicting fragment poses in a way that takes into account the molecular geometry and leverages Newton-Euler equations in rigid body mechanics, making it more effective than simply predicting an invariant update in the local coordinate system. The neural network predicts forces on each fragment atom to compute the total force and torque on each fragment, which is more effective than predicting only the change in translation in the local frame.

Experimentally, the authors demonstrate that their proposed model can generate valid, synthetic-accessible, and low-energy molecules under both unconstrained and constrained generation settings, outperforming other existing generative models.

Overall, the paper presents a novel good-quality work with good motivation and theoretical justification. For experiments, LinkerNet achieves satisfying results on benchmark datasets compared with existing methods.

**Weaknesses:**

The experimental results are not reported with error bars (standard deviation), and the training complexity, training time, and sampling time are not compared between different methods.

**Questions:**

Question:
1. How is the training/sampling complexity compared with existing methods, especially with DiffLinker?

2. I would like to see the error bars (standard deviation) presented in the table results.


Typo:
1. Line 154, draw samples and computer $\rightarrow$ draw samples and compute.

---

> ### Author Rebuttal · Authors · 2023-08-09
>
> We thank the reviewer for the feedback and suggestions. Please see below for our responses to the comments.
>
> **Q1: How is the training/sampling complexity compared with existing methods, especially with DiffLinker?**
>
> A1: For the training complexity, DiffLinker converges within 300 epochs and takes 76 hrs with one V100 GPU as the original paper reported. Our model converges within 50 epochs and takes 48 hrs with the same type of GPU. For the sampling complexity, DiffLinker finished sampling linkers for 43 PROTAC fragment pairs (100 linkers for each pair) in 132 min while our model takes 761 min with the same NVIDIA 1080 GPU. We thank the reviewer's suggestion and will add the training/sampling complexity analysis in the updated manuscript.
>
> **Q2: I would like to see the error bars (standard deviation) presented in the table results.**
>
> A2: We have computed the error bars for the main results by running the sampling procedure 3 times with different random seeds. The results are shown in Table 1/2 in the general response.

---

> > ### Comment · Reviewer_4AKh · 2023-08-14
> > **Response by Reviewer**
> >
> > Overall I find this paper well-motivated and properly justified. I will keep my score for a weak accept.

---

### Official Review · Reviewer_aUNA · 2023-07-04

**Soundness:** 3 good
**Presentation:** 3 good
**Contribution:** 2 fair
**Rating:** 6
**Confidence:** 4

**Summary:**

The paper presents a diffusion model for molecular linker design. Given the 3D structures of two molecular fragments, the model generates a pose for each of the fragments, as well as types and positions of atoms that can be added to link the two fragments in a single molecule.

**Strengths:**

The method and evaluation seem sound.

On the whole, the work and its relation to prior works are clearly explained.

If you are interested in designing PROTACs, then it is a nice practical contribution.

**Weaknesses:**

In section 3.3 I found the description of the fragment pose prediction module hard to understand. I didn’t understand the description leading up to equation (12) and I would like to know the architecture of the neural net used for fragment pose prediction. Is it another GNN the same as the linker GNN?


**Questions:**

‘in scenarios involving new drug targets… relative position between fragments may not be readily available’ – why is this the case for new targets and not older ones?

How much freedom is there really in relative fragment poses for PROTACs? It would be nice to see real pairs of examples with same heads, different relative poses, and both having high experimentally verified bioactivity. In line 274 I have no idea if the stddev of 0.1 fragment distance is realistic.

Line 83 ‘this is always the case’ should be ‘this is not always the case’

Line 120: What are ‘atom features’ as in ‘number of atom features’?

Line 116: say what Na, Nb, NF are first, then give the other definitions that use them.

Why do linker bond types need to be explicitly generated? Can’t they be inferred from linker atom positions?

Did you experiment with other samplers besides the one described in equation (2)?

How did you decide on the noise schedules for all the different things – fragment rotation, linker atom types, linker atom positions and so on?

Line 147 and line 154 should say ‘compute’, not ‘computer’.

Line 178: what is V_M under the summation? All the atoms in the molecule?

Is the ‘baselines’ comparison, where fragment poses are randomly corrupted, fair to DiffLinker etc.? If the fragments are presented in a relative pose for which no sensible linker can be designed, then not only does this give DiffLinker an impossible task, it also presents DiffLinker with an input that is presumably out-of-distribution relative to the partially obscured real molecules that it was trained on.

Are there always 2 fragments? Do you sample position and rotation for both? You could fix one without loss of generality.

Line 258 and table 1: is ‘Recovery rate’ how frequently a single sample is identical to the reference, or is it how often the reference appears in the list of 250 or 100 generated samples?

Line 321 and Table 4: is row (a) of table 4 just the same as row ‘ours’ of table 1, except that one is for ZINC and the other for PROTAC-DB?

Line 326: ‘liker’ should be ‘linker’.

Why is the fragment pose prediction module separate? Why not use a single GNN to predict ‘forces’ on all fragment and linker atoms, with a special readout head to do the update in equation (16)?

**Limitations:**

The methodology seems very specific to this problem.

---

> ### Author Rebuttal · Authors · 2023-08-09
>
> We thank the reviewer for the detailed feedback and suggestions. Please see below for our responses to the comments.
>
> **Q1: “In section 3.3 I found the description of the fragment pose prediction module hard to understand...”**
>
> A1: Equation (12) describes a way to update poses by predicting the pose change and applying it to the current pose, i.e. $(R_t, p_t) \circ (R_{t \rightarrow 0}, p_{t \rightarrow 0}) = (R_t R_{t \rightarrow 0}, p_t + R_t p_{t \rightarrow 0})$, which is same as the formula in Supplementary 1.8 Structure Module in the AlphaFold2 paper [Jumper et al., 2021]. The neural network predicts the rotation and translation change $R_{t \rightarrow 0} = \phi_R $ and $p_{t \rightarrow 0} = \phi_p$, which are invariant to the global rigid transformation, and apply them to the current noisy rotation and translation $R_t$ and $p_t$ to obtain the denoised ones $\hat{R}_0$ and $\hat{p}_0$. We have added more explanation before introducing Equation (12). It’s also worth noting that this updated formula is suboptimal and is *not* the one we used. We compare this way with our proposed physics-inspired pose prediction module in Table 2 to show the superiority of our method.
>
> The fragment pose prediction module is one graph attention layer built upon the linker GNN, instead of a separate GNN. As Figure 2 shows, we added one graph attention layer ($\phi_f$) on the top of $h^L$ to predict the force and torque. We have clarified it in the updated manuscript.
>
> **Q2: “‘in scenarios involving new drug targets… relative position between fragments may not be readily available’ – why is this the case for new targets and not older ones?”**
>
> A2: “Involving new drug targets” means the molecular fragments are bound with two protein targets instead of one. In this case, the relative position between fragments becomes unknown since the protein-protein binding pose has flexibility (see the example Fig.1 in the general response).  We have clarified this point in the revised manuscript.
>
> **Q3: “How much freedom is there really in relative fragment poses for PROTACs?”**
>
> A3: We find two PROTACs whose fragments bind with the same protein target (BRD4) and E3 ligase (CRBN) as Fig.1 in the general response shows. The linkers differ with a ‘COC’ motif. As we can see, the fragment poses differ a lot even with a slightly different linker.
>
> **Q4: “Why do linker bond types need to be explicitly generated? Can’t they be inferred from linker atom positions?”**
>
> A4: The heuristic approach to infer bond types is sensitive to the choices of hyperparameters and atom positions (the bond distance distribution is very sharp), and is not always reliable. Thus, we choose to predict bond types simultaneously.
>
> **Q5: “Did you experiment with other samplers besides the one described in equation (2)?”**
>
> A5: No, this is the standard choice of the diffusion process since we can directly draw samples from $q(x_t | x_0)$ and compute the posterior in closed-form.
>
> **Q6: “How did you decide on the noise schedules for all the different things – fragment rotation, linker atom types, linker atom positions and so on?”**
>
> A6: We choose to use the cosine schedule [Nichol and Dhariwal, 2021] with s=0.01 as all noise schedules. This is a common choice in DDPM. We follow this way and find it works well empirically.
>
> **Q7: “Is the ‘baselines’ comparison, where fragment poses are randomly corrupted, fair to DiffLinker etc.?”**
>
> A7: In the constrained generation setting, we randomly sample fragment poses but also make sure the candidate anchors in two fragments are facing to each other and there is no clash between fragments. This setting has been close to the real scenario for generative models which can not predict fragment poses, otherwise most of the other generative models would all fail.
>
> **Q8: “Are there always 2 fragments? Do you sample position and rotation for both? You could fix one without loss of generality.”**
>
> A8: Great question! Yes, PROTAC always involve two proteins and thus two binding molecular fragments. Although fixing one and sampling the other is equivalent to sampling two poses mathematically, it will involve the fussy coordinate system transformation for linker atoms and may make the learning process more difficult. For instance, assume the current ligand pose is [-M1+] [L] [-M2+] and the true pose is [+M2-][L][-M1+], where M1, M2 are two fragments, L is the linker and +/- denotes the two sides of fragments. The model would have to move all linker atoms in L and M2 to the other side if we fix M1, but it could simply mirror rotate M1 to achieve  [+M1-] [L] [-M2+] (same as  [-M1+] [L] [-M2+]) if we sample poses for both fragments. In preliminary experiments, we test sampling one relative pos / sampling fragment distance and two rotations, but neither of them is as effective as sampling two poses.
>
> **Q9: Line 258 and table 1: the definition of ‘Recovery rate’**
>
> A9: We sample 250 or 100 linkers for each fragment pair in the test set. If any linker is same to the reference linker, we count it as recovered for this fragment pair. The recovery rate is averaged over the whole test set.
>
> **Q10: Line 321 and Table 4: is row (a) of table 4 just the same as row ‘ours’ of table 1, except that one is for ZINC and the other for PROTAC-DB?**
>
> A10: Yes. The molecules in PROTAC-DB are much larger than the ones in ZINC. Without appropriate constrained guidance, the energy is easy to be very large. Another reason is that we applied the hard bond mask based on the candidate anchor sets as described in Sec 3.4.  Without guided sampling, the model may generate strange bond angles leading to high energy.
>
> We thank the reviewer for pointing typos out! We will fix them in the updated manuscript.
>
>
> **References:**
> * John Jumper et al. Highly accurate protein structure prediction with alphafold. Nature, 2021.
> * Alexander Quinn Nichol and Prafulla Dhariwal. Improved denoising diffusion probabilistic models. ICML, 2021.

---

> > ### Comment · Reviewer_aUNA · 2023-08-16
> > **Thank you for your clear response.**
> >
> > I have read the reply and looked at the PDF and my questions are resolved.

---

### Official Review · Reviewer_kzw2 · 2023-07-10

**Soundness:** 3 good
**Presentation:** 2 fair
**Contribution:** 3 good
**Rating:** 7
**Confidence:** 4

**Summary:**

The authors describe a novel method for the computational design of linkers using equivariant diffusion models. This is coupled with a fragment pose prediction step that allows the design of linkers without first knowing the relative orientation of the fragments.

**Strengths:**

- Comprehensive related works section
- Novel and sensible approach, allows for constrained generation
- Good empirical results support the paper's claims
- Good ablation experiments for the pose prediction module

**Weaknesses:**

- The authors need to better introduce the problem for a more general machine learning conference. Though well-defined for biologists, concepts like ubiquitination need some explanation for the NeurIPS audience.
- The writing needs to be improved. The authors need to explain the equation forms of their method, to give an intuitive sense of what their method does. This is especially critical between lines 149-164.

**Questions:**

- In line 92, please clarify why this is suboptimal compared to your approach. This should be changed in the text to also be used as a point of comparison with your method.
- Is there any data or evidence to back up the claim in line 280? Otherwise, this is a big limitation of the method.

**Limitations:**

The authors have mentioned limitations in the conclusion, albeit with a short sentence. This should be expanded upon.

---

> ### Author Rebuttal · Authors · 2023-08-09
>
> We thank the reviewer for the feedback and suggestions. Please see below for our responses to the comments.
>
> **Q1: “The authors need to better introduce the problem for a more general machine learning conference… The writing needs to be improved … ”**
>
> A1: We thank the reviewer for the suggestions on the writing part.  We have updated our manuscript by adding more explanations of the biological terminologies for the more general ML audience and adding more descriptions between equations.
>
> **Q2: “In line 92, please clarify why this is suboptimal compared to your approach.”**
>
> A2: The SMILES representation is not an optimal choice since it fails to capture molecular similarities and suffers from the validity issue during the generation phase. A small modification of the molecule formula will significantly change the SMILES string. These drawbacks are clearly discussed in [Jin et al., 2018] and we have added more explanation in the Related Work.
>
> **Q3: “Is there any data or evidence to back up the claim in line 280?”**
>
> A3: In the original paper of 3Dlinker and DiffLinker where the fragment poses are fixed, the novelty and uniqueness for their approaches are 29% / 32% and 24% / 30% on ZINC, which are way lower than the results of 50%+ / 90%+  in our new setting. In addition, the energies of molecules generated by 3DLinker and DiffLinker in our new setting are high, indicating their molecules are less stable. This evidence indicates that the actual chemical linker design space to form a low-energy molecule is small, supporting our claim in line 280 and related results.
>
>
> **Reference:**
>
> Wengong Jin, Regina Barzilay, and Tommi Jaakkola. Junction tree variational autoencoder for molecular graph generation. In International Conference on Machine Learning, pages 2323– 2332. PMLR, 2018.

---

> > ### Comment · Reviewer_kzw2 · 2023-08-14
> > **Increased score**
> >
> > Thank you for your responses. Since most of my criticism was due to writing and this has been addressed, I have increased the score of my review.

---

### Author Rebuttal · Authors · 2023-08-09

We thank all reviewers for their efforts and time in evaluating our submission and providing valuable suggestions and feedback.  In the general response pdf, we

* Add one example to show that the fragment poses are **not** fixed in the PROTAC design (Figure 1). Both PROTACs bind with the same protein target and E3 ligase, but the fragment poses and linkers are different.

* Add error bars for the main results in ZINC and PROTAC-DB datasets (Table 1, Table 2). The results still support our claims in the main text.

Please let us know if there are any additional concerns we can address for you to consider raising your initial rating, thanks!

---

> ### Comment · Area_Chair_Fric · 2023-08-13
>
> Dear Reviewers,
>
> Thank you for reviewing this paper. Authors have provided their rebuttal. Would you please check it, and give your comments/rating based on the rebuttal letter and the comments from other reviewers?
>
> Best Regards
>
> AC

---

### Decision · Program_Chairs · 2023-09-21

**Decision:**

Accept (spotlight)

**Comment:**

This paper introduces a new and interesting method for the computational design of linkers by using equivariant diffusion models. All reviewers are positive on this paper, and they point out that the method is novel and experiments are extensive. The method is also effective as shown in the experiments. There are some minor weaknesses in the paper, such as writing, which however can be addressed in the final version. AC agrees with the reviewers, and believes this work is worth publishing in NeurIPS.